# Characteristics of High-Resource Health System Users in Rural and Remote Regions: A Scoping Review

**DOI:** 10.3390/ijerph20075385

**Published:** 2023-04-04

**Authors:** Michele LeBlanc, Tomoko McGaughey, Paul A. Peters

**Affiliations:** 1School of Nursing, University of British Columbia, Vancouver, BC V6T 1Z4, Canada; 2Department of Health Sciences, Carleton University, Ottawa, ON K1S 5B6, Canada

**Keywords:** health care, rural health, healthcare inequalities, public health systems research, health care costs, health services research

## Abstract

A small proportion of health care users are recognized to use a significantly higher proportion of health system resources, largely due to systemic, inequitable access and disproportionate health burdens. These high-resource health system users are routinely characterized as older, with multiple comorbidities, and reduced access to adequate health care. Geographic trends also emerge, with more rural and isolated regions demonstrating higher rates of high-resource use than others. Despite known geographical discrepancies in health care access and outcomes, health policy and research initiatives remain focused on urban population centers. To alleviate mounting health system pressure from high-resource users, their characteristics must be better understood within the context in which i arises. To examine this, a scoping review was conducted to provide an overview of characteristics of high-resource users in rural and remote communities in Canada and Australia. In total, 21 papers were included in the review. Using qualitative thematic coding, primary findings characterized rural high-resource users as those of an older age; with increased comorbid conditions and condition severity; lower socioeconomic status; and elevated risk behaviors.

## 1. Introduction

A large body of research has identified that a small proportion of the population often accounts for a disproportionately larger proportion of utilized public health system resources [1,2,3]. Much of this usage is related to systemic, inequitable access and disproportionate health burdens faced by these individuals [2]. While high resource use impacts the wellbeing of high-resource users, it also depletes scarce health care resources and increases health service expenditures [1]. Thus, addressing the unmet needs of high-resource users is integral in ameliorating the efficacy of health care systems [3].

Within the literature, this broad group of health system users is identified by a variety of definitions and constructs, and includes high-cost, high-frequency, repeat, or high-resource users [4,5]. Research includes examination of unplanned hospitalizations, emergency department visits, or physician usage, includes a range of conditions, and encompasses individuals across the life-course. Despite the many definitions, these studies are similar in their focus on those individuals who use greater than expected proportions of often-limited health system resources [4].

Broadly, high-resource users are categorized in the literature as older aged adults from lower income quintiles with poorer self-reported health, often presenting with multiple chronic conditions and reduced access to primary health care [1,2,3,4,5]. Other social determinants of health have also been related to high-resource users, including unemployment, limited social support, rates of chronic illness, and rates of serious psychological illness and addiction [2,6]. Additionally, geographic differences have been identified, with certain areas demonstrating higher rates of high-resource use than others [5]. These factors, as well as other social determinants of health, often underpin or exacerbate frequent use, largely due to increased rates of unmet health needs, poorly managed chronic diseases, and comorbid conditions [6,7,8]. Characteristics associated with high-resource users tend to differ depending upon the focus of the study, often fluctuating depending upon distinct regions, services, or populations [2]. These findings indicate that an interplay of individual-level, community-level, and geographic factors contribute to differential rates of health-seeking behavior [7].

Rural and remote communities demonstrate persistent health inequities in comparison to their urban counterparts, largely due to geographic and structural inequities [7,8,9,10,11]. Individuals living remotely often report burdensome travel times and associated time and cost expenditure as drawbacks to accessing health care [11]. Common challenges faced by rural and remote communities—notably a transient workforce, fragmentation of service providers, and resource shortages—compound geographic barriers in health care access. Community-level difficulty in recruiting and retaining physicians also reduces service access, as limited available local services impede early identification and treatment of health concerns [7].

Despite structural barriers in health care, very few studies explicitly focus on high-resource users between and within rural regions, and no systematic or scoping review has been performed for this population. With an overwhelming majority of urban-centric studies, gaps in rural-focused research limit our understanding of factors that impact a community’s and/or an individual’s susceptibility to being (or becoming) a high-resource health care user in rural communities. 

Although rural areas are highly heterogeneous, rurality is often considered as a singular factor when considering social determinants of health [12]. To account for the heterogeneity of rural communities, we must better understand the determinants within and between rural areas [13]. The challenges faced in determining characteristics of high-resource users in rural communities are compounded by a lack of solidarity in the definitions of rural and high-resource. Such heterogeneity creates barriers in understanding and categorizing users that place a disproportionate burden on the health care system.

### Objectives

As illustrated, there is no universal understanding of the individual factors that are associated with high-resource users in rural regions, and how these factors interact. The objective of this review was two-fold: (1) to examine characteristics of high-resource users between and within rural communities in Canada through a rural-centered lens, and (2) to identify gaps in current research regarding the effect of rurality on being (or becoming) a high-resource user. Results from this review can thus be used to better inform future research and interventions aiming to address high-resource users in the health care system. 

A secondary objective of this research was to inform planners and practitioners in countries with public health systems and large rural and remote populations. While several countries could be included, our focus was on Australia and Canada, two similar countries with geographically vast rural landscapes, similar issues in rural health service provision, and long traditions of rural health research. We opted to limit our inclusion to these countries, as with the inclusion of others, it may have been difficult to disentangle differences between national health systems, thus diluting conclusions for rural health service provision.

## 2. Materials and Methods

A structured scoping review was undertaken to identify and to analyze characteristics of high-resource health care users in rural Canadian and Australian communities. This review was guided by the work of Arksey and O’Malley [14] and followed PRISMA-ScR guidelines on scoping reviews [15]. Following the review, a thematic analysis was undertaken to synthesize the literature and identify key themes. Findings were then collated, summarized, and reported.

### 2.1. Key Terms

The term high-resource user encompasses several terminologies and categorizations that describe the population using a disproportionately higher than expected level of health care services. Common definitions include high-cost, high-frequency, and repeat users, along with categorizations such as avoidable hospitalizations, less-urgent hospital presentations, and cumulative risk of readmission. For the remainder of this paper, the term high resource will be used to denote all users that would fall into any of these categories.

Researchers categorize rurality using different definitions and indexes, depending upon geographical location, region, or focus. For this paper, the term rural will be used to denote populations or health care regions that are categorized as such by researchers, regardless of the parameters of their definition. The working definitions included in the review will be provided and analyzed.

For this paper, the term Aboriginal will be used as a general term inclusive of First Nations, Inuit, and Métis people of Canada, in addition to Aboriginal and Torres Straight Islanders of Australia. 

### 2.2. Identifying Relevant Studies 

First, a search for rural high-resource users was conducted in PubMed, Web of Science, and Scopus (see Appendix A). Publications were limited to research on populations in rural Australia and Canada, published from 2000 to 2022 and in English. Second, results were combined, and duplicates removed. Third, two researchers conducted a title and abstract screening process of the combined results. Criteria for exclusion at the this stage were not focusing on characteristics of high-resource users or on Australian or Canadian populations, focusing on a single disease, or not including analysis of a rural population. Fourth, the authors conducted a full-text review with the same inclusion criteria as those for the title and abstract review.

### 2.3. Classification of Data

Thematic coding was conducted, and studies were categorized to identify the key characteristics of high-resource users. The primary themes were extracted and organized based upon individual- and community-level factors. Theme headings for individual-level factors included age, sex, socioeconomic status (SES), risk behaviors, Indigenous status, and presence or absence of primary care providers. Community-level themes included degree of rurality and degree of access to health care.

### 2.4. Strength of Evidence

We included a strength of evidence analysis to assist in guiding recommendations for future studies on this population. Papers were categorized by method employed, sample size, population sample, consideration of social inequities, inclusion or recognition of Indigenous populations, and inclusion of sex and gender. We further evaluated studies for depth of analysis, which considered whether the above categories were included and addressed in the study.

## 3. Results

The primary search produced 5528 results in Scopus, 460 results in PubMed, and 116 results in Web of Science, for a total of 5742 after duplicates were removed, as shown in Figure 1. Titles and abstracts were reviewed and assessed for eligibility based on the inclusion criteria, resulting in 64 articles deemed eligible. Following full-text review, 21 articles were selected for inclusion in this synthesis, summarized in Appendix A.

### 3.1. Individual-Level Characteristics

Individual-level characteristics were extracted from included articles and are summarized below in Table 1.

#### 3.1.1. Older Age

Almost half of the studies included in the review identified older age as a risk factor for high-resource use of health care services. Older age was associated with greater ICU and hospital admittance [16,17,18,19,20,21,22]; longer hospital stays [17,22,23,24]; and higher usage of primary care [17,22,25] and specialist services [17]. Additionally, those aged 65 years and older accounted for more than 50% of high-resource users in remote Australia [26] and for more than 50% of high-resource users of general health services in Ontario, Canada [27]. 

#### 3.1.2. Sex

Most studies reported sex as a predictor of high-resource use, with females more likely to be high-resource users than males. Sex-based differences were dependent upon other factors, however; females were identified as higher resource users in multiple health care settings, including health services for all ages [16,17,21,22,28], health services for youth [29,30], primary care for all ages, and adult emergency department presentations [21,31]. 

Penning (2016) found that sex-based differences were dependent upon which health service was being accessed [17]. Of people who accessed primary care services at least once, females made more visits; however, of those who accessed specialist services or were hospitalized, males made more specialist visits and had higher hospitalization rates. Women also had longer hospitalizations [17,20,23,32]. 

Manos (2014) examined sex-based differences in a population aged 12–24 years in Nova Scotia, Canada. Females accounted for double the primary care and inpatient contacts and made up 84% of all high-resource users compared to males [29].

#### 3.1.3. Comorbidities 

Individuals with higher comorbidity scores or indices—i.e., the Charlson Comorbidity Index (CCI)—are more likely to have higher comorbidity burdens [16,17,21,22,28,32]. Patients with higher scores on the comorbidity indices were more likely to be hospitalized [28] and to seek primary care and specialist visits [17]. One study found that a higher comorbidity score (CCI) nearly quadrupled the odds of having four or more admissions and that a decrease in wellness score increased the risk of frequent admittance by 53% [27]. 

Studies also identified significant comorbidity burdens within their respective high-resource cohorts. Guilcher (2016) found that 58.5% of high-resource users presented with eight or more distinct comorbid conditions, while less than 1% of the cohort had no pre-existing condition [27]. Garne (2009) identified three or more primary care diagnoses among almost 50% of high-resource patients [26], while Quilty (2019) determined that 47% of the high-resource cohort presented with three or more comorbid conditions [33].

#### 3.1.4. Socioeconomic Status

Many of the included studies identified measures of SES as important characteristics of high resource use, where low SES increased the likelihood of being a high-resource user [20,21,23,28,29,30,31]. Wallar (2020) found that lower income was one of the largest risk factors for high resource use for both males and females [20]. Nearly one quarter of participants in the analysis conducted by Longman (2012) reported difficulty affording their medications; one third of participants were using five or more medications at the time [23].

Some studies determined that the influence of high versus low income [17] or high versus low SES [29] varied according to the health service accessed. Penning (2016) determined that lower to middle income groups were significantly less likely to access primary care and specialist services compared to the highest income group [17]. However, among those who accessed primary care services at least once, those with lower income demonstrated a greater increase in use over time [17]. Additionally, among those that were hospitalized during the study period, the two lowest income quintiles had significantly higher hospitalization rates compared to the highest quintile [17]. 

Social isolation and psychological distress were also shown to be common among high-resource users in some studies, where a significant proportion lived alone and had lower social network scores [23]. The proportion of high-resource users that lived alone was significantly higher than in the general population, at a rate of 34% compared to 25%, respectively [23,32]. Additionally, 22% who reported needing help to care for themselves did not have a close friend or relative that could regularly care for them [23].

#### 3.1.5. Risk Behaviors

Risk behaviors, including smoking and excessive alcohol consumption, were identified as predictors of high resource use in 10 of the included studies [16,18,20,22,28,31,32,33,34,36]. Ten studies identified excessive alcohol consumption, and five studies identified smoking. One study found that heavy smoking was one of the largest risk factors regardless of sex [20], and another found that alcohol was the largest risk factor [18]. Springer (2017) found that the odds of being a frequent user were higher among those with a single alcohol-related condition; individuals with two or more alcohol-related hospital episodes had a higher risk of being a frequent user for more than one year. 

#### 3.1.6. Aboriginal Populations

Of the 21 papers included in the review, seven included Aboriginal populations [18,20,26,28,33,34,35,36]. Of these, five studies identified an increased risk of high health care use in Aboriginal populations. The remaining two papers did not discuss differences between Aboriginal and non-Aboriginal presentations. Springer (2017) found that Aboriginal patients were twice as likely to be frequent users (21.7%) in comparison to non-Aboriginal patients (10%); the two groups also had different clinical presentations. Risk of prolonged high resource use was elevated for Aboriginal patients [18], as was risk of hospitalization [28].

#### 3.1.7. Other Factors

Other factors were identified as predictors of high resource use, including being unmarried or widowed [19,20], homeless [29,35], physically inactive [20], food insecure [33], and underweight [19]. Wallar (2020) found that being underweight was one of the largest risk factors for high resource use [20]. High-resource users were also more likely to be infrequent or non-users of clinics, indicating unmet primary health needs [26].

### 3.2. Geographic Factors

#### 3.2.1. Rural Definitions

All of the papers included in this review discussed geographic indicators of high resource use, per inclusion criteria. Of these, 11 papers found that rural patients were more likely to be high-resource users, two found service use depended upon which service was being accessed, and one paper reported that urban patients accounted for more high-resource users. The remaining seven papers only analyzed rural populations and, therefore, did not provide a rural/urban comparison. 

Ten studies used various indices to classify rurality. Three used the Accessibility Remoteness Index of Australia [24,28,35], one used the Victorian health service regions [28], one used the Rurality Index of Ontario [27], and two used the Statistics Canada Rurality Index through the Canadian Community Health Survey [17,20]. The indicators used, their definitions, and the articles that used them have been provided in Table 1. Dufour (2020) and Chiu (2022) used a self-generated (not referenced) categorization based upon area populations: metropolitan area ≥100,000 inhabitants; small town: 10,000–100,000 inhabitants; and rural: <10,000 inhabitants. 

The remaining 11 studies did not specify how rurality was classified. Seven studies treated rurality as a dichotomous variable (rural or urban); four classified it based upon home postal code [20,21,22,32], while the other three did not indicate a categorization method [18,18,30]. Four studies did not provide definitions of rural versus urban, but all took place in preestablished rural regions [25,26,33,34]. Two studies did not specify rural definitions [23,36]. These definitions are summarized in Table 2.

#### 3.2.2. Health Service Use

Rural patients posed a larger burden in terms of rates for ambulatory care-sensitive conditions (ACSC) [16,20,35] and general emergency department attendance [21,22,25,30,32,33] compared to patients in urban centers. Quilty (2019) found that 85% of frequent attenders came from communities in more remote locations. Penning (2016) found that rural frequent attenders were more likely to be hospitalized, whereas urban folk were more likely to draw on general practitioner and specialist services in general. Another study indicated that hospitalization rates, average length of hospital stay, total length of hospital stay, and readmission rates all increased with increasing distance that a patient lived from the hospital [36]. 

In contrast, two studies determined that urban populations comprised an increased proportion of high-resource users. While Springer (2017) only identified a slight increase in urban frequent use than in rural, Guilcher (2016) found that urban dwellers accounted for almost 90% of high-cost health users [18,27].

#### 3.2.3. Service Access

Five studies analyzed the impact of service access on high-frequency use; all five of these identified low access/remoteness as a risk factor [19,23,24,33,35]. Greater access problems were associated with higher admission rates in all ACSC categories [35] and in hospitalization rates in general [19]. In one study, people who reported inadequate access to health care services had 10 times higher hospitalization rates than those with the highest service access [19]. Of those, people who lacked access to a primary care doctor demonstrated an 8% increase in hospitalization rates and 11–14% higher admittance rates than baseline [19]. Inadequate access to hospital services resulted in twice the raw probability of admission compared to that of the baseline group, with a 25% increase in admissions for physical conditions and a 21% increase in admissions for mental conditions [19]. Inability to fully access hospital services raised emergency department visits by more than 20% for both mental and physical conditions [19]. 

Two studies identified inaccessible transportation as a predictor of high-frequency use. Quilty (2019) noted that only 20% of high-frequency users had a car and, due to rural and remote living, no public transport was available [33]. Similarly, Longman (2012) found nearly one third of high-frequency users did not have a car they could drive and, of those living alone, 29% did not have access to a car at all [23].

## 4. Discussion

### 4.1. Individual-Level Characteristics

From this review, several individual-level characteristics were identified as key risk factors for high resource use and for how often services were accessed and which services were accessed most frequently. 

#### 4.1.1. Older Age

The relationship between age and health care expenditure has been of growing interest to rural health researchers and policy makers alike due to the impending health care implications of rapid population aging [37]. While older age in and of itself was identified as a risk factor for high resource use, Longman (2012) identified other aspects of social deprivation, including lower social network scores and social isolation, among older high-resource users [23]. Interventions to ameliorate health outcomes of aging Canadians should consider the social implications of aging, including social isolation, due to the associations between social networks and health outcomes.

#### 4.1.2. Sex

Although most of the papers in this review analyzed sex-based differences, none considered the social implications of gender. The ramifications of entrenched gender norms, especially in rural and remote communities, largely impact health-seeking and health behaviors. These place men at greater risk of undetected disease and subsequent illness exacerbation in comparison to females. 

The relationship between sex and high resource use was less linear, as sex served as an indicator of which type of service was more frequently accessed. For example, females were more likely to access preventative, primary, and specialist care services, whereas males were more likely to be hospitalized. In this case, it can be seen as a positive factor for hospitalization that more females seek preventative care.

Generally, men are less likely to partake in preventative health and to seek health care. Prior studies suggest that men in rural communities are especially vulnerable, as they are less likely to use preventative health services than their urban counterparts [26]. Men are also more likely to partake in risk behaviors, such as excessive alcohol consumption and smoking. 

Current gender-neutral or gender-blind research may be failing to identify social factors that impact other risk factors for high resource use. These social risk factors could serve as targets for intervention, which could prevent dependence on health care services by ameliorating baseline levels of health.

#### 4.1.3. Comorbidities

People with higher comorbidity indices face greater risk of illness exacerbation and complication, thereby increasing care use. Most high-resource users with comorbidities would benefit from multidisciplinary primary and/or specialist care, suggesting that high-resource users have unmet health needs and/or poorly managed chronic diseases [26].

In terms of quantifying comorbidities, Penning (2016) argues that a broader set of health status indicators would be more appropriate when examining whole populations [17]. For example, inclusion of acute conditions may better reflect the health care needs of younger populations who may not have developed comorbid conditions yet. This would indicate an underestimate of comorbid conditions in younger populations, thereby suggesting greater comorbidity burdens than reported.

#### 4.1.4. Socioeconomic Status

In the case of rural and remote regions, generalized lower SES, lower education attainment, and increased health burdens place rural health seekers at further disadvantage. While rural and remote regions face higher comorbidity burdens in comparison to urban centers, interventions such as multidisciplinary care are often not practically feasible due to prevailing inadequacy in rural and remote health care services, in which limited resources and staff shortages challenge health care delivery. 

Despite universal care rebates and the potential for private health insurance coverage, lower SES was a risk factor for lower access to care [17]. The inability of lower income groups to access adequate care contributes to a growing, negative cycle: lower SES contributes to poorer health, which increases the need for money spent on health care needs, which then worsens SES [29].

#### 4.1.5. Risk Behaviors

Excessive alcohol consumption and smoking were both identified as risk factors for high resource use. Excessive alcohol consumption was identified as the largest risk factor for high resource use in the Northern Territory of Australia [18], whereas smoking was one of the largest risk factors for adults living in Québec [20]. Risk behaviors are elevated both in rural and remote communities and in Aboriginal communities, thereby placing these populations at greater risk for related comorbidities and related outcomes. Springer (2017) highlighted the associations between alcohol and other comorbidities, highlighting its risk of exacerbating existing susceptibility to high resource use and its impact on others.

#### 4.1.6. Aboriginal Status

Aboriginal and non-Aboriginal people demonstrate distinct presentations of high resource use. High-resource users who were Aboriginal were found to be younger and more commonly female than their non-Aboriginal counterparts. Such differences were largely attributed to considerable social disadvantages for Aboriginal populations, which serve to exacerbate existing inequities in health outcomes and access to health care [33]. 

Historical oppression and marginalization of Aboriginal people propagate widespread disengagement from mainstream health care systems. Aboriginal populations face barriers in accessing western health care due to pervasive colonial disadvantage. Common concerns include fear of discrimination, lack of trust or confidence in providers, and culturally inappropriate service [28]. To improve the health outcomes of Aboriginal populations, the systemic barriers and inequalities must first be addressed through broad policy and local practice initiatives [28]. 

### 4.2. Geographic-Level Characteristics

#### 4.2.1. Rural and Remote Determinants of Health

The literature included in this review cited rural and remote living as a risk factor for high resource use. Despite commonly referring to rurality as a risk factor for high resource use, limited studies discussed the implications of this discrepancy across multiple conceptions of rurality [12]. Of those that did discuss geographic determinants, decreased access to health care and challenges in maintaining a stable work force contributed to high resource use [23,24,25,27]. 

Despite longstanding geographically based health discrepancies, the role of “place” in shaping health behaviors has not been clearly defined [38]. This may be due to prior research that determined sociocultural characteristics were relatively unimportant in explaining health care service use [38]. The characteristics of the people who use health care services are still the focus of most high resource use studies, rather than the communities from which they are sought [38]. This is despite evidence that health services can be in part shaped by the communities in which they are located [39]. In studies on high resource use, place is most often delineated as urban or rural and ignores both the manner by which “rural” can be defined and the significant heterogeneity between rural communities. 

#### 4.2.2. Access to Health Care

Inadequate health care services in rural and remote communities can enable undertreated or untreated health conditions, thereby increasing the risk of escalating existing conditions [19]. People from rural and remote locations tend to have longer hospital stays, which is partially attributed to lower access to care [24]. For example, when a doctor suspects a patient has little to no access to ambulatory care, they may be more inclined to keep the patient for observation [36]. Similarly, they may more readily admit patients if there is a possibility of serious complications with insufficient local resources. This scenario would be similar for a patient that does not have adequate transportation to or from a hospital. Lack of transport also exacerbates social isolation, which was identified as a risk factor for high resource use in elderly populations. 

Often, challenges in recruiting and retaining physicians widens gaps in health services in rural and remote communities. In addition to a lack of specialists and subspecialists [26], rural regions also struggle with primary care practitioner shortages [25]. Lack of health services leads to challenges in accessing care, often due to long travel distances and/or inaccessible transportation options. Even if general practitioners are available in the region, patients are less likely to seek care if they perceive the doctor as unavailable [19]. Availability of primary care doctors is integral in preventing hospital admissions, especially in rural and remote regions [28], as suboptimal provisions of local health care continue to contribute to excessive emergency department burdens [19]. 

### 4.3. Health System Consequences of High Resource Use

Although high-resource health care users account for a disproportionate amount of health care resources used, they are often overlooked in the literature and in rural health policy discourse. This broad-based scoping review produced 21 studies, with each demonstrating the extent to which high-resource users impact local/regional health systems. However, no national-level review has been conducted. To create a framework to ameliorate the health of high-resource users, nationally funded studies are needed. Such data can then enable evidence-based alterations to health policy, especially in relation to better serving those at risk for high resource use via prevention and primary care.

#### 4.3.1. Prolonged High Resource Use

Generally, few studies investigated long-term patterns of high-resource users. Differentiating between short-term and long-term frequent attenders could enable identification of characteristics to create more specific, targeted interventions for prolonged high-resource users. Prior research suggests that persistent high-resource users may benefit from case management and increased continuity of primary care in comparison to temporary high-resource users [40]. Like the neglect of heterogeneity in rural populations, the vast variation between and within high-resource cohorts is markedly overlooked, thereby undermining more targeted efforts to ameliorate high health care resource use.

#### 4.3.2. Length of Follow-Up

Longitudinal studies are currently lacking in the literature; these could enable the examination of whether and how relationships between people and health care services have changed over time [17,20]. In the analysis by Penning (2016), researchers found that declines in access to hospital care are coupled with increasing income-related disparities in access to care. Such trends would denote increasing inequities in access to health care, thereby challenging the equity principles in publicly funded health care systems [20]. 

In addition to larger-scale studies that compare health use within and across provinces, identifying localized areas of need would enable targeting of resources for populations as opposed to individuals [28]. This would be especially applicable in rural and remote populations, due to community-wide health determinants. 

#### 4.3.3. Length of Hospital Stays

Many studies focus on avoidable hospital visits (i.e., ACSC) to measure high resource use; however, studies indicate that lengthy hospital visits contribute more to emergency department congestion [40]. Supporting this, Springer (2017) determined that the duration of high resource use was elevated among older adults compared to younger high-resource users, indicating prolonged elevated health utilization and expenditure. 

### 4.4. Literature Gaps and Potential for Further Studies

Overall, the strength of high resource health research in Canada and Australia is low, due to narrow geographic breadth, lack of longitudinal analyses, exclusion of population groups, and limited consideration of social determinants of health. Although most studies included in the review identified remoteness as an indicator of high resource use, few discussed the rural-specific factors that promote such dependence upon health care services. 

#### 4.4.1. Social Determinants of Health

The primary social inequities considered in the included studies were education, income, employment, and homelessness. Three studies did not include consideration of social inequities, whereas the remaining 18 included social inequities in analyses, discussion, or both. Ansari (2006) acknowledged the inability to include social determinants of health due to inaccessibility of data. Wallar (2020) analyzed and discussed the impacts of immigrant status and race in high resource use patterns, where he found that immigrant status was a protective factor. He attributed this to the healthy immigrant effect [20]. No other studies considered the impact of immigrant status or race. 

#### 4.4.2. Consistent Rural Definitions

Despite diverse realities within and between rural communities, geographic determinants are often operationalized as urban or rural [38]. Such dichotomization of rurality denotes limited understanding of the heterogeneity of rural populations and masks disparities between and within rural regions. Evaluations of geographic disparities in health outcomes are difficult due to the multiplicity of ways in which rural has been defined [41]. 

While researchers and policy makers alike have attempted to create a standardized definition of rural, many assert that one definition would be insufficient due to the heterogeneity between rural communities [41]. Although the various definitions of rural serve as a shortcoming of individual studies, the discrepancies point to widespread systemic barriers that impede rural-centric classifications on national and international levels. 

#### 4.4.3. Consistent High Resource Definitions

High-resource users were also treated as a homogenous group, despite differing health needs and risk factors based upon a person’s sex, age, and geographic location. Definitions of high resource use varied depending upon study design and region. To address high resource use on a national scale, standardized methods of quantifying high-resource users must be developed and adopted, with consideration of differences based upon geographic location. 

### 4.5. Study Limitations

Scoping reviews are limited by an inherent risk of selection bias, as relevant sources may have been omitted due to relevance to the research question. To mitigate bias, the scoping review was completed by two researchers with predetermined exclusion and inclusion criteria. 

The disparities in definitions of rural and high-resource between papers also serves as a limitation on a review level. Although this was accounted for in the discussion, the lack of standardized terminology challenges the collation of findings. 

Further, due to a limited pool of Canadian literature on rural high-resource users, studies in Australia and Canada were grouped and analyzed together. This analysis may have overlooked the differences between Australian and Canadian health infrastructure and inequities, i.e., in relation to Aboriginal populations. The research question was limited to the study of high-resource users in Australia and Canada; and as such, the results will have to be tested and possibly verified in other rural and remote regions.

## 5. Conclusions

Despite promises of universal health care, a significant proportion of the Canadian population struggles in accessing health care and in staying healthy, which subsequently promotes high resource reliance on health care services. Shortcomings in providing adequate health care reveal significant gaps within the current national health care infrastructure. To alleviate the economic and health burdens of high-resource users, the risk factors of being or becoming a high-resource user should be identified and subsequently targeted in mitigation efforts. This review served to identify risk factors of rural high resource use based upon the available literature in Canada and Australia. 

The primary risk factors for being or becoming a high-resource user as identified in this review include older age, female sex, increased need factors, lower SES, elevated risk behavior(s), and Aboriginal status. However, differing levels of risk were identified between and within these categories. For example, although high-resource users were more commonly female than male, males were more at risk of frequent hospitalization. 

Although these findings summarize those of the available literature, national-level studies are necessary to better understand high resource use on a national level and subsequently alleviate its burden on the national health care system. Once individual risk factors for high resource use are identified, geographic determinants of high resource use should be identified to provide interventions where they are most needed. Indeed, the selected studies did not provide insight on community or societal-level impacts of high resource use. Further, the structural health care shortfalls that propagate high resource use would need to be identified and addressed.

## Figures and Tables

**Figure 1 ijerph-20-05385-f001:**
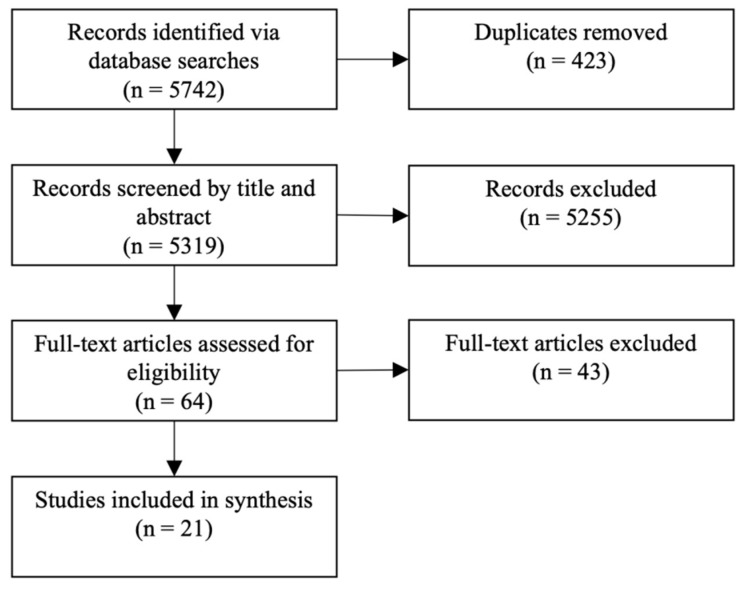
Scoping flow diagram.

**Table 1 ijerph-20-05385-t001:** Summary of individual and community-level findings.

Themes	Summary	Citations
Age	Those in older age groups more likely to be high-resource users, especially females	[16,17,18,19,20,21,22,23,24,25,26,27]
Sex	Females are more likely to be high-resource users and have higher rates of health service utilization compared to males	[16,17,21,22,28,29,30,31,32]
Comorbidities	Increasing risk of high resource use with increased number and complexity of comorbidities	[16,17,21,22,27,28,32,33]
SES	Lower SES increased the likelihood of being a high-resource user, with income the largest risk factor	[20,21,23,28,29,30,31]
Risk Behaviors	Smoking and regular alcohol consumption are strong risk factors for high resource use	[16,18,20,22,28,31,32,33,34,35,36]
Aboriginal	Higher prevalence of high-resource users among Aboriginal population, although this is confounded by income and lower health service access	[18,20,26,28,33,34,36]
Self-Rated Health	Those with poorer self-rated health and self-reported mental health are more at risk of being a high-resource user	[16,18,20,22,28,31,32,33,35,36]
Rurality	Rural and remote communities have higher proportions of high-resource users	[16,17,19,20,21,22,23,24,25,26,28,29,30,31,32,33,34,35,36]
Health Service Use	Frequent service users were more likely to be from rural or remote communities	[16,20,21,22,25,30,32,33,35]
Service Access	More high-resource users in areas with lower access to primary health services	[19,23,24,33,35]

**Table 2 ijerph-20-05385-t002:** Definitions of rurality in included literature.

Indicator	Definition	Example
Accessibility Remoteness Index of Australia (ARIA)	Outer Regional: ARIA score greater than 2.40 to ≤5.92; significantly restricted accessibility to goods, services and opportunities for social interaction. Remote: ARIA score greater than 5.92 to ≤10.53; very restricted accessibility to goods, services and opportunities for social interaction. Very Remote: ARIA score greater than 10.53 to ≤15; very little accessibility to goods, services and opportunities for social interaction.	Ansari (2006) [35] Brameld (2006) [24]
Victorian DHS Regions	Corresponds to ARIA definitions. 13 metropolitan areas: ARIA scores below 0.50. 19 rural areas: ARIA scores above 0.90.	Ansari (2013) [28]
Rurality Index of Ontario	Scaled index based on population factors and distance. Communities with values ≥ 40 are considered rural.	Guilcher (2016) [27]
Statistics Canada Statistical Area Classification	Categorization based on Statistics Canada population, and accessibility definitions based on if the respondent’s residing census subdivision is a census metropolitan area (urban), a census agglomeration (urban), or a census metropolitan influence zone (3 degrees—rural).	Penning (2016) [17] Wallar (2020) [20]

## Data Availability

Data extraction instrument is available upon request from corresponding author.

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
