# Peer review of "Characteristics of High-Resource Health System Users in Rural and Remote Regions: A Scoping Review"

_ijerph, 2023, doi:10.3390/ijerph20075385_

Round 1
Reviewer 1 Report
This paper focuses on an understudied topic and delivers substantive findings. It is well-written and structured, and I have only a few minor concerns before this paper is ready for publication. Although these concerns are minor, I do encourage the authors to address them in full.
1. I am not quite sure about what are the pre-determined selection criteria for filtering the literature from 225 to 73, as stated in lines 152-153. It is important for the authors to convincingly articulate how the two researchers select the samples, what criteria did they use, and how are the criteria applied.
2. The authors discussed the gender difference, and they stated that "females were more likely to access preventative, primary, and specialist care services, whereas males were more likely to be hospitalized" in lines 353-354, while "Prior studies suggest that men in rural communities are especially vulnerable, as they are less likely to use preventative health services than their urban counterparts" in line 395-396. What are the authors' opinions on this? For females, can their high-resource use in primary and specialty services be understood as a positive factor for avoiding hospitalization? Females tend to use healthcare resources more because they are less healthy, or they just care more about their health status on a daily basis?
3. In line 475, the authors stated that place is understudied. This is not accurate. There are many studies on the physical and psychological associations between place and well-being, particularly for the elderly. I would recommend the authors rephrase this statement - is it the case for the current literature, or only the case for your scoped research?
4. in Section 4.3.1, I am surprised that these studies only discussed the individual-level impact of prolonged high-resource use. What about the cost to society? Does the prolonged use generate a crowding-out effect for others? What do the authors think about this?
5. A more important ethical debate related to my last question is that, do you think high-resource use is justified for the certain populations specified in this paper? From an equity perspective, those who need care more should have more access to care. Then what is the purpose of studying high-resource use after all? This justification is something missing in the introduction, which I think would be interesting for the authors to discuss since it will highlight the significance or contribution of this paper.
Reviewer 2 Report
The article proposed for my review is entitled "Characteristics of High-Resource Health System Users in Rural and Remote Regions: A Scoping Review".
This article wishes to focus on the case of patients considered as high resource health system users, especially in rural and remote regions. This research objective stems from the observation that very few studies explicitly focus on high-resource health system users between and within rural areas, and no systematic or scoping reviews have been conducted for this population. In this article, the authors propose to analyze the individual factors that contribute to the use of resource-intensive health services in rural areas.
To do this, the authors conduct a scoping review to identify and analyze the characteristics of high-resource health system users in rural Canadian communities to also identify the effect of rurality on high-resource health system users.
The article is interesting. The research design is coherent and the research objectives seem relevant to me, although they need to be somewhat clarified. On the other hand, certain elements (described below) seem to us to require further study or additional information from the authors:
1. The authors' definition of rurality seems ambiguous or not sufficiently developed. The authors indicate that researchers give different definitions of rurality but they indicate that they only use the term rural to conduct their scoping review. Would it not be possible to specify what type of rurality the article is about by describing more precisely the Canadian (and Australian) geographical situation in terms of health or socio-economic conditions? This would perhaps allow the title and research objective of the article to be clarified by specifying which rurality we are talking about, since there are multiple forms of rurality in the world (desired or suffered ruralities, rich or precarious economic ruralities, etc., community or non-community ruralities, etc.).
2. The authors state that they conducted their scoping review on high-resource health system users in rural Canadian communities. However, they later state that they limited their scoping review to Australia and Canada because these countries have similar health infrastructures and geographies. A clearer justification for these similarities in terms of geography and health infrastructure, but also population in rural areas, should be provided. Otherwise, it is difficult for the reader to understand why articles on Australia are selected to analyze the characteristics of high users of health resources in rural communities in Canada, as stated in the research objective. Indeed, it is conceivable that other territories in the world (e.g., Brazil, Mexico, even parts of the United States, etc.) may have geographic or indigenous population characteristics similar to Canada. Why the choice of Australia to illustrate the Canadian case?
3. There is no mention by the authors of the keyword equation that was used to conduct the literature review in PubMed, Web of Science and Scopus.
4. The authors indicate that exclusion and inclusion were performed in several steps. In order to better inform the reader, it would be appropriate for the authors to mention in a table, for each of these steps, the exclusion or inclusion criteria that were used. Moreover, in Figure 1, at the end of a step, it is mentioned (Title exclusion (N = 340), isn't it rather Title inclusion?
5. The results of the scoping review put several individual factors of high resource health system users (Older age, Gender, Comorbidities, Socio-economic status, Risk behaviors, Aboriginal population). It would be interesting here to have a table describing precisely these individual factors (which age groups, which gender, which types of comorbidities, which type of economic difficulties, etc.). These factors could then be discussed further in the discussion section.
6. The authors note that the literature included in this review cites living in rural and remote areas as a risk factor for high resource use. They then describe the characteristics of this rurality in terms of the determinants of health in rural areas, access to health care, and physician recruitment and retention. These results are particularly interesting because they qualify rurality in terms of high health resource utilization. It would be appropriate to discuss them in more detail in the discussion section in order to relate them to the individual characteristics of high-resource health systel users.
7. The discussion of the results is interesting, but they focus only on certain characteristics of care or their territorial determinants (Prolonged high-resource use, length of follow-up, length of hospital stays, consistent definition of rural) without always associating them with the individual characteristics of the high-resource health system users (age, sex, comorbidities, risk behaviours, indigenous population) presented in the results section. This association could make it possible to present a synthesis table indicating both the individual factors of high-resource health system users and territorial factors related to rurality. In this way, it would be possible to respond in a more synthetic way to the two research objectives of the article, which were, I recall, "gathering information of the individual factors that contribute to high resource healthcare service use in the lens of the individual's residential rurality," and "informing planners and practitioners in countries with public health systems and large rural and remote populations. Therefore, the conclusion could be more concise and focus on these two types of results.
In view of a future publication of this article, I invite the authors of this article to take into consideration these (non-peremptory) suggestions for improvement.
I thank the authors for giving me the opportunity to read this article and wish them a future publication of their work.
Reviewer 3 Report
Thank you for the opportunity to review the manuscript entitled: Characteristics of High-Resource Health System Users in Rural 2 and Remote Regions: A Scoping ReviewThis is an interesting topic with a high quality and relevance.
I just have some comments to the authors.
Line 182: Please change the format of 2X to make it more readable for everyone.
Line 232: Please clarify what “SES-based” stands for. This is the first time described in the paper.
Line 237: Please change the format of 3X to make it more readable for everyone.
Line 294. I am not sure if this table is adding information to the main goal of this paper. I believe it could be mentioned in the body of the paper briefly, saving some words and space. I will defer it to the authors.
Congratulations for this interesting and thorough job
Reviewer 4 Report
The scoping review aims to gather information of the individual factors that contribute to high resource healthcare service use in rural areas individuals, to better inform future research and interventions aiming to address high-resource users in the health care system.
I would like to thank the authors for this work because this work could make contributions to understanding of factors that impact a community’s and/or an individual’s vulnerability of being a high-resource health care user in rural communities, but I believe that the work done has important gaps in the methodology, which risks nullifying the important effort the authors have made.
Abstract
The abstract is a little wordy and can be simplified. The abstract should briefly state: Purpose, Methods, Results, Discussion, Conclusions.
Introduction
There is a lack of definition and references on the concepts of rural areas within public health systems and rurality as a social determinant of health (Lines 69-75)
Paragraph 1.1 is a bit vague. It is suggested that the authors be more direct about the objectives this specific research and expected results.
Methodology
The research methodology leaves too much room for individual assessment regarding the eligibility of the selected studies. Inclusion and exclusion criteria are not clearly defined. Why did authors used “Population, Intervention, Comparison, Outcomes and Study (PICOS)” checklist?
The search string is not reported. It is suggested to include a table with the search string and the keywords.
The methodology for analysing the strength of the evidence is not comprehensible.
What is the classification methods?
In general, the methodology is too wordy. It is suggested that the authors briefly report, point by point, all the concrete steps that enabled the selection of studies.
Results
The order of the PRISMA steps has not been respected.
Figure 1 is confusing. How many articles were included during the screening. If the eligibility criteria were not reported, how were the articles selected according to the full test?
Duplicates should be eliminated during the identification phase.
Without the search string, the accuracy of the reported data cannot be verified.
The results should be reported in two tables according to the classification of the data, described in section 2.3
Discussion
The results and discussion are redundant. It is suggested that the results be reported in table form and analysed in the discussion.
Round 2
Reviewer 2 Report
I thank the authors of this article for their efforts to make their article more intelligible. The research objectives of this article are now clearly stated and fairly convincing details of the methodological choices have been provided. The results present more precise contributions. However, we still note three points that could be improved and allow the presentation of these results to be better structured:
1. Section 4.1. presents the individual characteristics of the high RESOURCE users mentioned in the literature included in the study. Section 4.2. presents the geographical characteristics of the rural areas with high resource use. Nevertheless, section 4.3, Strength of Evidence Analysis, is not well structured. It includes information on the consequences of high resource users on the health system (Prolonged high-resource use, Length of follow-up, Length of hospital stays). This section 4.3. could be entitled "Consequences of high resource users on the health system".)
2. sections 4.3.4 - 4.3.5 - 4.3.6 present the limitations of the literature studied and the prospects for new studies on high resource users. This section could be entitled 4.4. "Limitations of the literature and prospects for further studies" with a presentation of section 4.4.1 - Social determinants of health - 4.4.2 Consistent rural definitions - 4.4.3 Consistent high resource definitions. Therefore section 4.4. "Study limitations" will be entitled 4.5.
3. in this last section 4.5. "study limitations", it should again be pointed out that the research question has limited the study of high resource users to Canada and Australia, the results of which will have to be tested and possibly verified in other rural and remote areas.
I thank you in advance for taking these last suggestions into account.
Author Response
Thank you for your continued helpful suggestions. Please find attached the point-by-point response where we have accepted all suggestions.

Reviewer 4 Report
The authors revised the manuscript according to my suggestions.
Author Response
Thank you for your helpful suggestions.